# Student Disaffection: The Contribution of Greek In-service Kindergarten Teachers in Engaging Each Preschooler in Learning

**DOI:** 10.3390/bs10020051

**Published:** 2020-02-05

**Authors:** Maria Sakellariou, Efthymia Tsiara

**Affiliations:** 1Department of Early Childhood Education, School of Pedagogical Sciences, University of Ioannina, 45221 Ioannina, Greece; marisak@uoi.gr; 2Department of Pre-school Education, University of Ioannina, 45445 Ioannina, Greece

**Keywords:** student disaffection, academic achievement, self-efficacy, abilities and personality development, classroom cohesion, personal commitment, educational equality, effective socialisation, active citizenship

## Abstract

Engaging each student in learning comprises a continuous challenge and concern for the contemporary teacher. Educational research confirms the alarming increase of the disengaged students, relating student disaffection to adverse effects on students’ academic development. In the present research through one-on-one, semi-structured interviews, we investigate 80 Greek in-service kindergarten teachers’ opinions with regards to the significance of engaging the disengaged students in learning activities in preschool environments. The interviews based on Creswell’s (2009) interview model, incorporate open-ended and close-ended questions that offer a well-rounded view of the subject. Qualitative and quantitative data analysis of teachers’ opinions show that engaging each disengaged preschooler has multiple benefits on students’ academic development, class climate, and cohesion, and teacher’s self-efficacy, as well. Specifically, teachers’ engaging actions offer students the opportunity to develop their abilities, self-efficacy, and sense of belonging. The interviewees also recognise that increased student engagement levels decisively affect teachers—students’ interactions, offering at the same time clear feedback to the teacher.

## 1. Introduction

School disaffection, as a social phenomenon has taken concerning dimensions. Educators and research communities have become preoccupied with increasingly high levels of student “detachment” from learning [1,2,3]. In-service teachers strongly emphasise the growing complexity of contemporary students’ social, emotional, and behavioural difficulties and the higher demands placed upon them with regards to effective dealing with everyday issues within classroom context [4].

Despite the alarming increase of the disengaged students, and the growing concern within the research community, the conceptualisation of student disaffection remains insufficient [5]. In general, the concept of student disaffection closely correlates with the multidimensional construct of student engagement [6]. The following definition offered by Skinner and Belmont indicates that conceptualisation of school disaffection bases on the opposite concept of student engagement:
‘The opposite of engagement is disaffection. Disaffected children are passive, do not try hard, and give up easily in the face of challenges … [they can] be bored, depressed, anxious, or even angry about their presence in the classroom; they can withdraw from learning opportunities or even rebellious towards teachers and classmates. Engagement versus disaffection encompasses the typical behavioural and emotional constructs from most theories of achievement and intrinsic motivation.’[7]

Furthermore, the constructs of student engagement and disaffection have always been keystones in theories of motivation [8,9]. Motivational conceptualisations of disaffection include behavioural, cognitive, and emotional components. According to Skinner and Pitzer [10], behavioural manifestations/indices of student disaffection comprise the ways that students withdraw from learning tasks, display inattention, passivity, as well as their mental counterparts (e.g., apathy or amotivation) and emotional reactions (e.g., frustration). The motivational model holds that disaffection is the result of unsupportive, even rejecting interpersonal interactions or perspectives of self as uncongenial, unqualified, or pressured in school [11,12]. 

Student disaffection has reportedly lasting and adverse effects on students [13,14]. The disaffected students risk low academic performance [6,12,15] and so, feel ineffective and marginalised having more possibilities to join disengaged peer groups [10,11,16]. When students experience school as unsupportive, constraining, and unfair, they have increased possibilities to become disaffected and drop out both figuratively (being present, but not taking part in tasks) or literally (early school leaving [13,17]). There is a great deal of evidence that dropping out of school is a gradual process that occurs over many years and often begins in elementary school [18]. 

Besides, student disaffection may undermine student-teacher mutual relations. In general, the quality of student-teacher relations contributes to developing either “cycles of student engagement” or “cycles of disaffection” [1,14,16,19,20]. Teachers’ emotional and instructional support is communicated to students and has a pervasive impact on the way in which students feel that their needs are met [4,21,22,23,24,25,26]. In fact, teacher’s support facilitates student engagement in learning [1,10,11,16,19,25]. This positive, motivating influence that teacher “launches” creates a cordial classroom atmosphere that draws students into learning, promotes their desire to learn [22,27], and fosters closer relationships [14]. Teachers, on their part, experience satisfactorily student engagement, and as a consequence, they tend to motivate and offer more support to the engaged students. ‘This is the so-called “virtuous cycle of engagement”, which makes motivational “rich” students richer and more engaged’ [10].

On the contrary, harmful interactions with classmates and teachers may consult in an endless cycle of false interpretations and mutual rejection. Teachers may experience student disaffection as a factor that hinders their teaching and undermines their self-efficacy [1,3,10]. Students’ unresponsiveness or misbehaviour may make teachers feel rejected, disturbed, and provoked realising that their efforts bring forth only partial or no results [4]. Students challenging behaviour may constitute a personal shortcoming and a significant source of anxiety and concern for the teachers and undermine their professional competence and efficacy.

Besides, students’ restless, disruptive, and provocative behaviour may result in increasing coercion from a teacher [1,3,4]. In case that teachers show coercion or indifference to already-disengaged students, neglecting their needs may reproduce disaffection, making matters worse [15,16]. Skinner and Pitzer (10) refer to a “vicious cycle of disaffection”. Unsurprisingly, teachers’ unsupportive reactions may render motivationally “poor” students more disengaged or gradually at-risk. However, whether teachers experience students’ disaffection as a diagnostic tool signalizing that the specific student needs more instructional and emotional support, teachers’ reactions may have a positive impact on him/her [10]. Additionally, if teachers make systematic use of teaching strategies that support emotionally and instruct the disengaged students, they may break the adverse circle of disaffection and foster re-engagement. 

## 2. Data Collection Methods

In reaction to the phenomenon of student disaffection and its consequent problems, research interest, and focus on students’ and teachers’ voice has arisen. The majority of the existing research focuses on students in Grade 6 and higher, during which there is a dramatic drop in student engagement levels [28]. Besides, even though student disaffection is inseparably related to drop-out and identified globally as a predictor to academic failure, in Greece, it has been given the attention needed. Educational research has focused on drop-out rates in Hellenic educational context, but has not qualitatively investigated those factors that decisively affect decline in student engagement. In addition, there is a notable lack of qualitative and quantitative investigation into in-service teachers’ opinions with regards to student disaffection and their contribution to combating the decline in classroom engagement. The teachers’ opinions have neither been investigated in any grade nor in preschool education, even though preschool environments can be critical to academic success and risk reduction of early school leaving [13,29]. 

Taken into consideration those mentioned above, we consider it interesting to investigate in-service kindergarten teachers’ opinions with regards to the importance of involving the disengaged students in learning activities. Therefore, our research lies in the hypothesis that in-service preschool education teachers recognise the multiple benefits of engaging each disengaged student and their contribution to disaffection decline. Specifically, the study addresses the following question: (a)how do teachers perceive the benefits of engaging each disengaged preschooler for the students himself, for the teacher, and the classroom cohesion?(b)do teachers consider student’s engagement levels to affect teacher-student relations?

The present research used semi-structured interviews as a data collection method. The interviews were designed according to Creswell’s interview model [30]. Adopting this model, during the research design, the researcher should determine the target study population and interview type and interview question types. In the phase of data collection, the researcher should locate a suitable and audio-friendly place for conducting each interview allowing for privacy and obtaining interviewee’s participation consent, as well. The researcher is also suggested to take brief notes and record each interview, securing an accurate record of the conversation, which is particularly important for coding in data analysis. Creswell [31] also suggests using probes for additional information, asking the interviewee to either clarify or elaborate. The final step in Creswell’s model is the interview closing, during which the researcher thanks and ensures the participants confidentiality. 

In our research, the interviews were based on an interview guide that included five parts investigating in-service teacher opinions pertaining to (a) student engagement components (47 items, questions 9–56); (b) student disaffection components (25 items, questions 57–82); (c) student engagement factors, (47 items, questions 83–130) (d) student engagement facilitating strategies (55 items, questions 131–186); and (e) student engagement assessment (11 items, questions 187–198). Each part combined open-ended and close-ended questions (Σ = 198), allowing the researcher more flexibility to explore interviewee’s opinions. The types of questions that the interviews incorporated were the background, knowledge, experience, opinion, feelings, and sense of gaining a rounded perspective [30]. 

This research started with a pilot study, which aimed to test and refine the data collection instrument (interview guide), and prepare the initial coding list, as well as the data analysis process. The pilot study ensured that the terminology of the questions was properly understood by the respondents. After the pilot, the main study took place (September 2017–May 2018). Each one-on-one interview lasted for about 50–60 min. The discussion was recorded—with the interviewee’s consent—and field notes were taken, avoiding the risk of losing data. 

Besides, interviews were based on predefined protocol allowing the researcher to structure the conversation flow. The protocol was intended only for interviewer use and not to be shared with interviewees. The protocol defined the interview duration (i.e., how much time is to be allotted to each interview guide part), the ground rules and probes, and interviewer introduction (researcher’s self-presentation, presentation of research purpose, and the reason that the interviewees have been asked to participate in the interview). In each interview, the conversation flow generally comprised the researcher’s introduction, the demographic information, and the 198 open-ended and close-ended questions.

In our research, we used different sampling methods. Convenience sampling was used in the pilot study, testing the interview guide on a convenience sample of people who had similarities to the target population for our study. During the main research, we used snowball sampling and benefited from one participant suggesting or introducing another interviewee to us.

The participants as a total were 80 Greek in-service preschool education teachers that were working as general (85%) and special (25%) education teachers in the prefecture of Ioannina and Larissa (Table 1). Besides, the vast majority of the kindergarten teachers (84.75%) had long teaching experience (more than 10 years), but have not advanced educational studies/qualifications, except for their basic studies (28.75% own a Master degree and 7.5% a doctoral degree). Teachers also participated in this study on a voluntary basis. 

## 3. Data Analysis Process 

Although there is no single approach to analysing qualitative data, the most important and agreed-upon guideline is that the analysis process is inductive and iterative, which is paramount to ensure authenticity [1,31]. Thus, in the present research, the coding process was iterative, since the code list, new codes that emerged from the data, and the adopted coding strategy were tested, improved, and confirmed. Specifically, an a-priori code list which was developed by the data from the pilot study and based on the conceptualisations of student engagement were elaborated with new codes emerging from the data analysis process in the main research [32]. 

The analysis occurred during transcribing (converting audio recordings into text data), memoing, and coding. After transcribing, we read the data thoroughly from beginning to the end without applying any codes, to develop an understanding of the whole data (a time-consuming process known as memoing). In the margins of transcriptions, we wrote our initial impressions (memos), which facilitated in searching for recurring themes [31]. These two analysis processes, in turn, led to coding, which was made up of the following three steps; open coding, (developing initial categories), axial coding (reconstructing the data in order to develop main categories and subcategories) and selective coding (demonstrating links and connections in categories) [30]. In other words, in the coding process, the a-priori code list was elaborated with new, frequently emerging codes. The three main categories (For the benefit of the class climate—class climate C.C.; For the benefit of the specific disengaged student—disengaged student abbreviation D.S; For the benefit of the teacher himself—teacher abbrev. T.) and subcategories (e.g., student’s self-efficacy, abbrev. S.S.E.) mentioned in the results section were developed through open and axial coding, while the selective coding demonstrated the connections among categories and subcategories. For example, when an interviewee reported: “*Each student affects the team…and the teacher”,* we used the codes (C.C) and (T) indicating that the specific participant recognises that the benefit of engaging each student is double and refers to the class climate and teacher, as well. Then, we further elaborated on the answer to specify the emerging dimensions (subcategories) and connected them with the three main categories and existing subcategories, or added new subcategories.

The quantitative analyses of the data (by 198 open-ended and close-ended questions) took place later using SPSS (IBM SPSS v.16.0) and Excel (Microsoft Excel 2007) in order to investigate if the data is affected by interviewees’ teaching experience and qualifications. However, in the present paper, we present a small part of the research data.

## 4. Results

### 4.1. Teachers’ Opinions with Regards to the Importance of Engaging the Disengaged Students in Learning Activities

In an attempt to demonstrate teachers’ contribution to engaging the disaffected student in preschool environments, in this paper, we present teachers’ responses to the following open-ended question (*Why is it important to engage each disengaged student?).* The following figure (Figure 1) depicts their answers that have been qualitatively and quantitatively analysed and are at first developed into three main categories (open coding and axial coding).
For the benefit of the class climate. Thirteen per cent of the interviewees considered it important to engage each disaffected preschooler aiming to foster and ensure classroom cohesion. ([Ν = 17, Σ = 128], Ν = The total of teachers’ responses in each category, Σ= The total of teachers’ responses.).For the benefit of the teacher himself. Twenty-seven per cent of the interviewees admit that their teaching efforts to engage the disaffected student have a pervasive impact on themselves. They recognize the importance of engaging the disaffected preschoolers, so as to experience self-efficacy considering their instruction effective, and their pedagogical objectives accomplished. The participant teachers also consider student engagement as a matter of ethics and personal commitment [Ν = 34, Σ = 128]. The extracts from the following interviews are indicative:
“… it is important for the teacher, to consider himself to be effective in teaching” Teacher 37, Head of school, Master in education (M.edu), <30 years (yrs) experience).
“This is our role, our responsibility: each student should “reach” a desirable (engagement) level.” (Teacher 28, Head of school, <20 yrs experience).
“… for the teacher to feel that he carries out his task.” (Teacher 41, <20 yrs experience).
“It’s a personal commitment of mine to help my students; … not only the disengaged ones, but also the class as a whole. Each student affects the team…and the teacher, as well.” (Teacher 56, <30 yrs experience).For the benefit of the specific disengaged student. The vast majority of the interviewees (60%) recognise as being of considerable significance to engage—first and foremost—each student who does not frequently take part in learning-related tasks, for his own benefit. Their efforts to involve each disaffected preschooler have a determinant impact on the target student [Ν = 77, Σ = 128]). 

After the first qualitative analysis of teachers’ responses at the specific open-ended question, a more meticulous analysis took place investigating the connections in the categories that have been developed by the “open” and the “axial coding” of the data (an analysis procedure described above as “selective coding”). The following figure (Figure 2) depicts teachers’ opinions with regards to the aspects of the disengaged student that benefit from teachers’ support and engaging teaching efforts. 

Specifically, the interviewees (Ν = 77) (The total of responses in the specific open-ended question according to the “open” and “axial coding” of the data) explain that their efforts to involve each disengaged preschooler in learning first and foremost focus on fostering and affecting the target-student at the following aspects: Abilities and personality development. Few teachers (12%) argue that their attempts to motivate disengaged preschoolers focus on offering them the opportunity to develop their abilities and unfold their personalities. Introverted, shy, hesitant students that frequently deprive themselves of expressing their needs and experiences need more support to unfold their abilities, talents, and character. (References in the interviews: 2, 16, 18, 39, 47, 57, 60, 72, 75 [Ν = 9, Σ = 77])Effective socialisation. Students who indicate disaffection characteristics are usually isolated or marginalised. According to 14% of teachers, these preschoolers that may have difficulty in communication, peer interaction, and cooperation need more opportunities to become effectively socialised and develop a sense of belonging as members of a class. Seemingly, these children have difficulty in dealing with interpersonal and school challenges. The interviewees also claim that their efforts to engage each disengaged focus on ameliorating teacher-student mutual relations, which in turn foster the learning process. (References in the interviews: 11, 16, 23, 25, 40, 42, 49, 50, 52, 65, 80 [Ν = 11, Σ = 77])Learning outcomes. Teacher’s efforts intentionally focus on the disengaged students providing them with more opportunities to get involved in tasks and consequently obtain learning outcomes. 16% of the interviewees argue that disengaged preschoolers need more instructional support since they usually spend insufficient time-on-task, delay or even fail to complete a task and have difficulty in meeting the requirements of school life. (References in the interviews: 1, 8, 14, 17, 22, 27, 41, 44, 56, 57, 63, 79 [Ν = 12, Σ = 77]).
“These students need our help! The self-regulated students learn on their own, while the disengaged student will not manage to do so if we do not support and guide them … without teachers’ scaffolding”. (Teacher 8: special education teacher, M. edu, <20 yrs experience)A positive attitude toward learning. Disaffected preschoolers are usually unwilling, half-hearted, or unenthusiastic. Disaffected preschoolers are not easily impressed and refuse to participate in organised classroom activities. Taking into consideration that indifference/apathy/amotivation constitute core manifestations of disaffection, 23% of teachers argue that their attempts to motivate the disengaged preschoolers to aim at developing a positive attitude toward learning. (References in the interviews: 7, 8, 9, 15, 32, 42, 49, 52, 58, 61, 65, 66, 67, 71, 74, 77/ [Ν = 16, Σ = 77]). The extracts from the following interviews are indicative:
“Teachers should serve a "starting point" for a change in the academic development of a disengaged student. This will foster his/her self-confidence and self-image among his classmates” (Teacher 52: M.edu, <20 yrs experience).Educational equality. Disengaged preschoolers usually have difficulty in meeting the requirements of school life, attributing their shortcomings to the reduced learning opportunities from their family environment. Taking that for granted, 18% of the interviewees argue that their teaching efforts intentionally focus on the target-students in order to provide them with equal learning opportunities. (References in the interviews: 7, 19, 20, 24, 27, 33, 35, 45, 53, 54, 57, 63, 67, 72 [Ν = 14, Σ = 77])
‘It is his right to learn, even though his progress is not significant’ (Teacher 27: <20 yrs experience)
‘It is important to give equal opportunities to each student. It is important to respect each child’s learning pace, to be interested in his progress, to adapt your expectations according to his needs. It is important to facilitate learning for the benefit of each student’ (Teacher 33: <10 yrs experience).Active citizenship. A few interviewees (6%) argue that disengaged students need increased opportunities to develop critical thinking and consequently take initiatives and actions, since they are described as being passive, indifferent and reluctant to participate not only in organised classroom activities but also in jointly shared actions (References in the interviews: 9, 39, 55, 57, 72/[Ν = 5/77])
I try to engage this specific student in learning so as to teach him to think independently, to decide, to take initiatives and as a consequence, make him an active member of the class initially and society later (Teacher 72: special education teacher, <5 yrs experience).Self-efficacy. 10% of teachers identify that their efforts to engage each disengaged student focus on fostering his/her self-efficacy. Disengaged preschoolers seem to be emotionally insecure, introverted, unconfident, and may need more teachers’ emotional and instructional support. (References in the interviews: 10, 29, 46, 52, 57, 59, 75, 80/[Ν = 8/77])
“A shy and introverted student needs the teacher’s encouragement.” (Teacher 10: <20 yrs experience)
“To encourage that student who may want, but due to his character hesitates to take part in a task. (Teacher 46: special education teacher, <5 yrs experience).

### 4.2. Teachers’ Opinions with Regards to the Impact of Student’s Engagement Levels on Teacher-Student Relations

In an attempt to demonstrate teachers’ contribution to engaging the disaffected student in preschool environments, we also present (Table 2) teachers’ responses to a relevant close-ended question: *(Teachers support and develop close and caring relations with each student, in accordance to his/her engagement levels*.) which can confirm the data presented above.

Almost half of the participants (45%) disagree with this question identifying that teacher’s support isn’t affected by student engagement level. They claim that they develop close and caring relations with each student, despite his/her engagement levels. In contrast, 42.5% of the participants admit that teachers’ support is affected by student engagement since this happens unconsciously. The following extracts indicate their ambiguous opinions with regards to the impact of student’s engagement on teacher-students’ relationships.
“Seemingly, the relationships with our students are affected by their engagement … but we try hard so as not to communicate it to them”. (Teacher 9: M. edu, <20 yrs experience)
“It is really encouraging seeing our students being engaged. Their engagement motivates us to continue, … while their disengagement discourages us”. (Teacher 11: M. edu, <20 yrs experience)
“If the teacher supports the engaged students more, does so unconsciously”. (Teacher 31: M. edu, <5 yrs experience).

Since interviewees’ responses to the specific closed-ended question are so ambiguous, we further used non-parametric tests to investigate if the data is affected by interviewees’ teaching experience and qualifications. The results of non-parametric tests show that the data from the closed-ended question do not record statistically significant differences in relation to teacher’s qualifications (Mann–Whitney U = 790,000; Z = −0.055; P = 0.957). Both qualified teachers and teachers who have no further training beyond their basic university education recognize that student’s engagement levels can generally affect and be affected by teacher-student relations. 

Furthermore, the results of non-parametric tests for two samples (1–10, > 10 years of experience) show that there are no statistically significant differences in the closed-ended question. Hence, teachers’ opinions are not differentiated by their teaching experience (Mann–Whitney U = 370,500; Z = −0.871; P = 0.384). Both teachers with long teaching experience and inexperienced ones recognize that the student’s engagement levels and teacher-student relations can be mutually affected.

## 5. Discussion–Conclusions 

According to the finding presented above, whether a disaffected student is engaged, not only does he/she benefit from his increased engagement level, and class cohesion, but also the teacher himself. Seemingly, these findings indicate that the participants are expressing their concerns about engaging each disaffected student. They identify that student disaffection may affect their personality and academic development. The learning process invites students to unfold various aspects of their character and develop their skills and talents. However, when a student refuses to participate, not only does he deprive himself of learning opportunities, but also does not reveal those traits of his personality and the way of thinking that renders him unique. 

Besides, the participants admit that student disaffection may affect student socialisation. The learning process, as a result of knowledge co-construction, invites students to coexist in the class respecting each other’s personality, exchanging experiences and knowledge, sharing feelings, working together and finally achieve learning outcomes. A disengaged student, however, loses the opportunities to collaborate with his classmates, to experience a sense of relatedness, and to enjoy the benefits of the learning process. In other words, in-service teachers recognise that student engagement levels affect students relationships with classmates and teacher, confirming the existing literature [3,16]. They also identify that disengaged students have increased opportunities to gradually become students at-risk confirming relevant educational research which indicates that school disaffection is the key indicator for predicting early school leaving [11,13,17]

In addition, the participants identify that motivating each disengaged student is of utmost importance. According to their reports, teachers consider student engagement as being the means for the education to provide equal learning opportunities and perform compensatory work. They recognise- confirming Parsons & Taylors’ (3) point of view- that engaging each disengaged student is a matter of justice since every student is entitled to have equal access to the educational system. 

Taking into consideration the findings presented above, we infer that the participant in-service preschool education teachers make sincere efforts to involve children in learning [33]. Their opinions should be considered as a reflection of the contemporary class, demonstrating some of the benefits and the difficulties of the regular class instruction. These findings point to the importance of learning to deal with disaffected students, which should form part of the training priorities of the preschool education teacher at this level.

In relevant research of the University of Crete in Greece, in-service teachers report that they do feel insufficiently prepared to manage challenging behaviours in the classroom and make the right choice and the most effective use of teaching techniques in each situation [4,34]. The participants express their need to be supported by school psychologists or counselors who are trained in educational issues and have experience of a collaborative partnership with education staff. International research also confirms the lack of experience of effective cooperation with and support from professionals [4]. 

Taking into account our findings in combination with relevant research findings, we conclude that in Hellenic educational context teachers’ engaging efforts focused on each disaffected student should be based on sufficient support by and constructive collaboration with the experts [35]. Teachers, along with the school counselor, school psychologist, even expert colleagues, are invited to collaborate, focusing on the target student and work on his emotions and experiences so that he will progressively engage in learning. Such collaboration with the experts is undeniably critical in preschool education so as to decline student disaffection and prevent early school leaving in upper Grades. For the time being, in the Hellenic educational preschool education, teacher-psychologist collaboration takes scarcely place and only in cases of engaging students with special needs or serious behavioural problems. 

To this end, future research should focus on exploring learner-centred strategies that evolve with each new generation of students so as to facilitate teachers to foster student engagement in either typical and special needs populations, or those at risk. 

In conclusion, increased and total student engagement level during regular class instruction constitutes a fundamental objective, if we expect quality in learning. Thus, one of the essential goals of teaching should be considered the promotion of active learning, which in turn contributes to developing lifelong learner identity and unfolding children’s personality that continues to change in meaningful ways throughout their lifespan.

## 6. Limitations

Interviews can provide a meticulous description of how teachers construct meaning about classroom engagement. However, interviewer’s knowledge skills, even biases, may affect quality, depth, and type of interviewees’ responses. There are also questions about the dependability and transferability of the interview findings [30]. 

Besides, different sampling methods used in the present research have their strengths and limitations, as well. While snowball sampling provides only slightly stronger results than does a convenience sample, and most research projects benefit from some amount of snowballing, this is a dangerous primary sampling design, as the risk inherent in snowball sampling is the overrepresentation of a single, networked group. 

As a consequence, we admit that the results of this research should be interpreted with caution given the small sample and be considered as the first step at the research level that aims to highlight important issues about student engagement. Further research is needed as an extension of this study.

## Figures and Tables

**Figure 1 behavsci-10-00051-f001:**
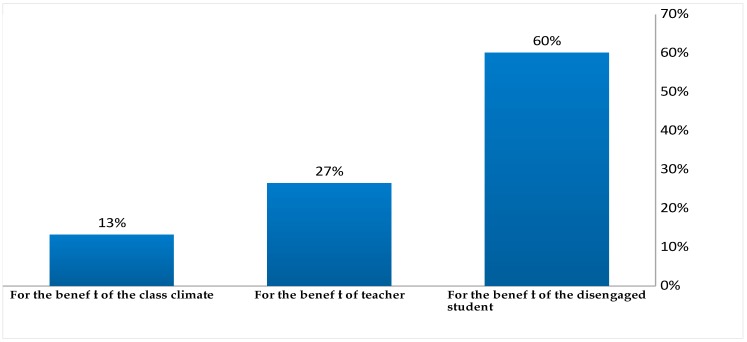
Teachers’ contribution to engaging each disaffected student in preschool environments.

**Figure 2 behavsci-10-00051-f002:**
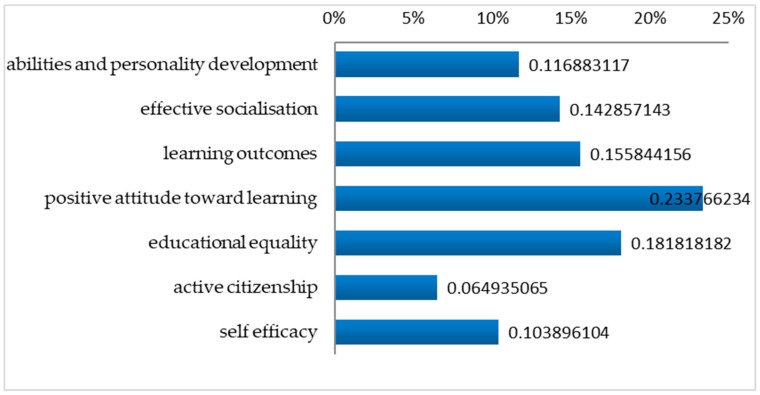
Teachers’ contribution to engaging each disengaged preschooler.

**Table 1 behavsci-10-00051-t001:** The participants of the research according to their working post, teaching experience, and qualifications.

Participants’ Working Post	Teaching Experience		Total	Qualifications		Total
	1–10 Years	>10 Years		Without Extra Qualifications	With Extra Qualifications	
General education teacher	4	42	46	25	21	46
Special education teacher	6	6	12	3	9	12
Head of the School	3	19	22	9	13	22
Total	13	67	80	37	43	80

**Table 2 behavsci-10-00051-t002:** Teacher’s support in relation to student engagement.

		Frequency	Percent	Valid Percent	Cumulative Percent	N	Valid	80
Valid	I absolutely disagree	15	18.75	18.75	18.75		Missing	0
I disagree	21	26.25	26.25	45	Mean	2.93	
I do not even disagree	10	12.5	12.5	57.5	Median	3.00
I agree	23	28.75	28.75	86.25	Mode	4
I totally agree	11	13.75	13.75	100	Std. Deviation	1.367
Total	80	100.0	100.0

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
