# Peer review of "Student Disaffection: The Contribution of Greek In-service Kindergarten Teachers in Engaging Each Preschooler in Learning"

_behavsci, 2020, doi:10.3390/bs10020051_

Round 1

Reviewer 1 Report

This is an interesting article, about an important aspect of education that is often neglected.

In general, the paper is sound. However, there are a few things that need to be rectified.

The English needs to be checked carefully. There are a lot of small mistakes and unidiomatic phrases. Please avoid contracted forms (isn't) and do not separate the subject from the verb by commas. Also, check all the possessive forms, use of articles, etc. Although the paper is generally well organised, I have some issues with the discussion section, because it goes beyond commenting on what the authors found in this part of their study. It is not ok to say, as they do,

"However, another concern arises. Are they adequately trained in teaching methods and strategies and assisted in engaging efficiently the disaffected students?..... teachers argue that their basic training is inadequate in preparing them to manage a disruptive form of student behaviour..."

The teachers may have said this in another part of the study, but they do not say it in the answers reported in this paper. So if the authors wish to discuss training implications, they should do so from a neutral perspective, saying that "These findings point to the importance of learning to deal with disaffected students, which should form part of the training priorities for teacher at this level...." Then they can go on to mention any bibliography about this. But I stress, it is not correct to "report" something stated by these teachers in part of the (larger) study as though it were something that has emerged from the present study.

Reviewer 2 Report

The manuscript entitled "The Contribution of Greek In-service Kindergarten 2 Teachers in Engaging Each Preschooler in Learning" aims to investigate kindergarten teachers’ perspectives with regards to the significance of engaging the 12 disengaged students in learning activities in preschool environments. 

Introduction

The researchers claim that 'There is a great deal of evidence that dropping out of school is a gradual process that occurs over many years and often begins in elementary school (line 56)'. However, there is no evidence provided that this is also valid in Greece context. It would make the claim stronger, if the researchers provide references fro the Greece context. 

Although the researchers stated that they investigate the teachers' perspectives, it appease that actually they collected and studied the opinions of the teachers, not the perspectives. This point needs attention, and should be corrected. 

Data collection

Did the researchers also collected quantitative data? This is unclear. 

It is mentioned that the researchers interviewed 80 teachers which is quite big number of population for a qualitative study. Some more information about these teachers, maybe a table presenting the participant teachers would be helpful. 

Data Analysis

The qualitative data analysis process was not expresses or done correctly and clearly. This part has some drawbacks. For example, in qualitative data analysis you cannot analyse the data in situ, but the data collection and data analysis is an iterative process. So, the researcher collects data, analyse the data, if it is needed (i.e. if the data is not saturated) the researcher collects more data, analyse the data, and so on. They also mentioned that 'The post analysis occurred during transcribing (line 116)'. This is not a correct approach if the researchers really did analysis in this way. You transcribe the all data, you read the transcribes throughly from beginning to end, without coding them, etc. There is quite important misunderstanding of qualitative data analysis. Moreover, in line 121 they have written that '...led to coding; the final step of data analysis.' The coming is NOT the final step of data analysis, in opposite it's one of the earlier stages. So. I suggest the researchers should learn how to analyse the qualitative data, and do analysis more appropriately. It would be helpful to read some articles as examples of this type of research. For example, 'Alonzo & Kim (2015). Declarative and dynamic pedagogical content knowledge as elicited through two video‐based interview methods', and 'Bayram-Jacobs, et al. (2019). Science teachers' pedagogical content knowledge development during enactment of socioscientific curriculum materials' can be good examples for the researchers to understand the qualitative data analysis and how to report it. 

Moreover, the code list is not provided and it not clear how the researchers developed the categories mentioned in the results. This process should be explained clearly.

Quality of Research

The quality of a qualitative research is explained through other quality criteria than quantitative research. The researchers talk about validity and reliability (qualitative criteria), it is more appropriate to mention criteria for qualitative research and discuss the research through these criteria. Read for example, Guba and Lincoln (1986). 

Results

The term 'perspectives'  should be reconsidered whether it's correct or better to use opinions. 

While presenting the results the researchers use '(References in the 139 interviews: 3, 5, 6, 7, 19, 21, 22, 24, 28, 33, 34, 35, 48, 56, 64, 69, 76)' apparently the researchers are from qualitative area, in qualitative research it's not necessary to include these in presenting the results. Instead the researchers could provide more quotations of the teachers to enrich the results section. That's more interesting to readers and the reference numbers. 

In the quotations provided the researchers mentioned the interview number, instead of this they can give more meaningful data for example, Teacher 1, 10 yrs experience, etc. 

In the results section, I would like to see the most important/significant results first. 

What is presented in figure 2 is not 'perspectives'. Please check this. 

Discussions/conclusions

In line 251 it is stated that 'Seemingly, these findings indicate that Greek in-service kindergarten teachers are expressing their 251 concerns on engaging each disaffected student.' This is qualitative study and you cannot generalise the results to whole Greek teachers. The results of this study is valid only for this group of teachers. The results maybe transferable to other setting but cannot be generalised. 

Conclusions did not provide me the take away message and the implications into the practice. 

Reviewer 3 Report

The theme is clearly defined and deals with a topic of interest such as The Contribution of Greek In-service Kindergarten Teachers in Engaging Each Preschooler in Learning. The 26 out of a total of 32 bibliographic references presented in the introduction, in support of the state of the art knowledge, mainly focus on the concept of "Student disaffection" (line 51), an otherwise welcomed concept that is not found either. in the keywords and not in the title of the research. A readiness of the reader to meet this concept would also be welcome. An express and less deductive presentation of the research objectives would also be welcome. Defining clearly measurable objectives would also allow the definition of hypotheses that would derive logically from the theoretical analysis and be oriented towards offering new solutions. Hypotheses can be formulated in theoretical but verifiable terms. Also, the description of the procedure at the level of detail that would allow the research to be reproduced would be welcome and would greatly increase the chances of the dissemination being cited.

Round 2

Reviewer 2 Report

The manuscript entitled "The Contribution of Greek In-service Kindergarten 2 Teachers in Engaging Each Preschooler in Learning" aims to investigate kindergarten teachers’ perspectives with regards to the significance of engaging the 12 disengaged students in learning activities in preschool environments.

Introduction

Regarding my first comment the authors provided a footnote on page 2, however it's not clear where/what this footnote refers. I still think that it's important to clarify the existence of the problem in Greek context. So,. it will be helpful to clarify this in the text instead of footnote. 

About the concept 'perspective': Perspective is different than just an opinion. I find it not suitable to use 'perspective' as a synonym of 'opinion'. Using 'opinion' would be more appropriate since the researchers asked the opinions of the participants , and did not study the 'perspectives'. 

Participants

It's important to provide detailed information about the participants of the research. I invite researchers to add this information. In the review process, if it was asked, you can go beyond the word limit. 

Data Analysis

In revising this part the authors have copied some statements exactly from the suggested articles of qualitative research. For example:

"the coding process was not linear but iterative where the code list, new codes, and coding strategy were checked, refined, and confirmed." (line 148-149)

"An a-priori code list was elaborated as a result of the pilot study." (line 150)

"the a-priori code list was updated with new codes that appeared to be significant and frequently emerged from the data." (line 160-161)

This is plagiarism and not allowed. Please paraphrase the sentences and write your own sentences. You also need to refer to the articles and you read and used for this part. 

The authors claimed that they did 'selective coding' as a third cycle of the coding process. However, it's not clear which themes arouse from the selective coding. 

In lines 167 & 168, the authors have written "we used the codes (C.C) and (T) indicating..."  they should make it clear what are C.C. and T? 

Results

For my comment: "What is presented in figure 2 is not 'perspectives'. Please check this." The authors replied:

"If the reviewer insists, we will replace the word, even though in Greek we would prefer the meaning of the word perspectives." 

The article they submitted is in english language so, the terminology should be chosen by considering this language choice. 

Reviewer 3 Report

The paper "Student disaffection: The Contribution of Greek in Kindergarten Teachers in Engaging Each Preschooler in Learning" addresses a current topic, the authors being able to capture the characteristics of this trend in education. Even if the research was conducted at the national level (Greece), we could say that this trend is found globally. Therefore, a detailed presentation of the research methodology would be welcome to allow for possible replication within other nations and, even later, even an international comparative analysis. The state of knowledge is presented in sufficient detail to understand the context. Further exploitation of the ability to use SPSS in this research would be welcome, as the data collected would allow refining of the results and a further development of the discussions and conclusions obtained.

Author Response

This manuscript is a resubmission of an earlier submission. The following is a list of the peer review reports and author responses from that submission.